# A Systematic Review and Meta-Analysis of the Pathology Underlying Aneurysm Enhancement on Vessel Wall Imaging

**DOI:** 10.3390/ijms25052700

**Published:** 2024-02-26

**Authors:** Ronneil Digpal, Kenton P. Arkill, Regan Doherty, Joseph Yates, Lorna K. Milne, Nicole Broomes, Orestis L. Katsamenis, Jason Macdonald, Adam Ditchfield, Ana Paula Narata, Angela Darekar, Roxana O. Carare, Mark Fabian, Ian Galea, Diederik Bulters

**Affiliations:** 1Department of Neurosurgery, Wessex Neurological Centre, University Hospital Southampton, Southampton SO16 6YD, UK; ronneildigpal@doctors.org.uk (R.D.); nicole.broomes2@nhs.net (N.B.); 2Biodiscovery Institute, University of Nottingham, Nottingham NG7 2RD, UK; kenton.arkill@nottingham.ac.uk (K.P.A.); msxlm9@exmail.nottingham.ac.uk (L.K.M.); 3Biomedical Imaging Unit, University of Southampton, Southampton SO16 6YD, UK; regan.doherty@uhs.nhs.uk; 4Department of Neuropathology, University Hospital Southampton, Southampton SO16 6YD, UK; joseph.yates@uhs.nhs.uk (J.Y.); mark.fabian@uhs.nhs.uk (M.F.); 5Faculty of Engineering and Physical Sciences, µ-VIS X-ray Imaging Centre, University of Southampton, Southampton SO16 6YD, UK; o.katsamenis@soton.ac.uk; 6Department of Neuroradiology, Wessex Neurological Centre, University Hospital Southampton, Southampton SO16 6YD, UK; jason.macdonald@uhs.nhs.uk (J.M.); adam.ditchfield@uhs.nhs.uk (A.D.); ana.narata@uhs.nhs.uk (A.P.N.); 7Medical Physics, University Hospital Southampton, Southampton SO16 6YD, UK; angela.darekar@uhs.nhs.uk; 8Clinical Neurosciences, Clinical and Experimental Sciences, Faculty of Medicine, University of Southampton, Southampton SO16 6YD, UK; r.o.carare@soton.ac.uk (R.O.C.); i.galea@soton.ac.uk (I.G.); 9Department of Neurology, Wessex Neurological Centre, University Hospital Southampton, Southampton SO16 6YD, UK

**Keywords:** intracranial aneurysm, magnetic resonance imaging, vessel wall imaging, pathology, histology, aneurysm-wall enhancement

## Abstract

Intracranial aneurysms are common, but only a minority rupture and cause subarachnoid haemorrhage, presenting a dilemma regarding which to treat. Vessel wall imaging (VWI) is a contrast-enhanced magnetic resonance imaging (MRI) technique used to identify unstable aneurysms. The pathological basis of MR enhancement of aneurysms is the subject of debate. This review synthesises the literature to determine the pathological basis of VWI enhancement. PubMed and Embase searches were performed for studies reporting VWI of intracranial aneurysms and their correlated histological analysis. The risk of bias was assessed. Calculations of interdependence, univariate and multivariate analysis were performed. Of 228 publications identified, 7 met the eligibility criteria. Individual aneurysm data were extracted for 72 out of a total of 81 aneurysms. Univariate analysis showed macrophage markers (CD68 and MPO, *p* = 0.001 and *p* = 0.002), endothelial cell markers (CD34 and CD31, *p* = 0.007 and *p* = 0.003), glycans (Alcian blue, *p* = 0.003) and wall thickness (*p* = 0.030) were positively associated with enhancement. Aneurysm enhancement therefore appears to be associated with inflammatory infiltrate and neovascularisation. However, all these markers are correlated with each other, and the literature is limited in terms of the numbers of aneurysms analysed and the parameters considered. The data are therefore insufficient to determine if these associations are independent of each other or of aneurysm size, wall thickness and rupture status. Thus, the cause of aneurysm-wall enhancement currently remains unknown.

## 1. Introduction

Intracranial aneurysms (IA) are common, with a prevalence of 3.2% in healthy people aged 50 [1]. They most frequently come to medical attention either when patients present with subarachnoid haemorrhage (SAH) after rupture or as incidental findings during imaging performed for other reasons. 

Vessel wall imaging (VWI) is a high-resolution contrast MRI technique that has been used to visualise the wall of IAs. Multiple cross-sectional studies have shown an association between contrast enhancement and unstable aneurysms manifesting with aneurysm growth or rupture [2,3,4,5,6,7,8]. VWI is therefore increasingly used in patients with SAH found to have multiple aneurysms in order to identify the culprit aneurysm requiring immediate treatment [9]. It is also a promising technique to stratify patients with unruptured intracranial aneurysms (UIA) into high- and low-risk groups and identify the need for treatment or closer follow-up [10].

Despite this potential, the pathological change underlying the enhancement seen on VWI is not clear. There are reported associations with aneurysm wall inflammation, endothelial cell integrity and vasa vasorum [11,12,13,14,15]. However, enhancement is also strongly correlated with aneurysm size, and it is unclear if any of these pathological findings are independently and causally related to enhancement. Many other putative mechanisms may underlie enhancement. Some reports demonstrate that slow blood flow along the aneurysm wall can result in a high-intensity rim similar to enhancement [16]. Partial volume effects could mean that thinner walls appear less intense than thicker ones, given that aneurysm wall thickness is typically smaller than the voxel size of VWI. Given that in some cases of avid enhancement, the area of enhancement can appear thicker than the wall itself, it is not necessarily clear whether the gadolinium is collecting within the aneurysm wall or the fluid and tissues surrounding it. Finally, although aneurysms that have already ruptured almost universally enhance, it is not clear whether these were all enhancing high-risk aneurysms before rupture or if rupture itself influences enhancement.

To provide some clarity regarding the pathological basis of enhancement, we performed a systematic review and meta-analysis of the histological findings in IAs with and without enhancement on VWI. Our aims were firstly to identify all histological parameters that have been reported in IAs that have been imaged with VWI; secondly, to use pooled data to study associations of histological parameters with enhancement; and thirdly, to perform multivariate analyses to identify independent histological predictors of enhancement.

## 2. Methods

This study was performed in accordance with the PRISMA [17] guidance. A PRISMA checklist is provided in the Appendix A. The protocol has not been published or registered.

### 2.1. Literature Search 

A search of the PubMed and Embase databases with no date restrictions, limited to articles in English for the terms “vessel wall imaging”, “MRI” and “pathology” (a full list of search terms is provided in Appendix A), was performed on 8 March 2021. Raayan QCRI software(App version 0.1.0) was used to evaluate the search. After using the “detect duplicates” functions to identify duplicated papers between the searches, all abstracts were screened by a single investigator (RD) for full-text review. After identifying initial publications, citations and studies citing the initial publications were searched; these were subsequently screened for full-text review. We included cohort studies, case–control studies and case series which reported on at least one histological marker from a saccular intracranial aneurysm wall obtained during surgery that had been imaged preoperatively with MRI vessel wall imaging at 1.5 or 3 Tesla and were written in English. Case reports, studies that reported on intracranial aneurysms thought to be of an immune-related aetiology, blister aneurysms and thrombosed aneurysms were excluded.

The bibliographies of each eligible study were searched at the initial screening phase for potentially eligible forward citations. Studies identified were found on Google Scholar (scholar.google.com) (accessed on 25 March 2021). (See Figure 1 for flowchart of studies).

### 2.2. Data Extraction

Data extraction was performed by a single investigator (RD). Data regarding aneurysm size, location, growth, rupture status and enhancement were extracted from the manuscripts and their supplements (See Table 1). Additional data were collected regarding imaging sequences and contrast used for the MRI VWI scan. Available pathological markers were recorded and are summarised in Table 2.

### 2.3. Analysis

Each study was assessed using either the Joanna Briggs Institute (JBI) checklist of case–control studies or case series and presented as risk-of-bias-assessment charts [18]. On the basis of the supplementary data provided by six of the papers, a one-stage individual-level aneurysm analysis of pathological findings and vessel-wall enhancement was performed. Initially, chi-square analysis and Cramer V association analyses were performed to determine the interdependence of the variables. A meta-analysis of associations for the 9 binary histological variables was calculated. The R package “Meta” was used to pool the data into a fixed-effects model and calculate odds ratios using the Mantel–Haenszel method with continuity correction to adjust for zero cells. Univariate followed by multivariate logistic regression was performed. All analysis was performed in R studio version 4.0.2 (22 June 2020).

**Table 1 ijms-25-02700-t001:** A summary of the studies included, including the proportion of aneurysms showing enhancement, the mean size of enhancing and non-enhancing aneurysms and patient demographics. Quan et al. [19] did not provide individual aneurysm data. Range depicted in brackets. Where the data is not available NA is shown.

	Year	Number of Aneurysms	Enhancing Aneurysms	Rupture Status	Size (mm)		Age (Years)	Female (%)
					Enhancing	Non-Enhancing		
Zhong et al. [20]	2021	27	19/27 (70.4%)	All unruptured	9.2 (7.6–11.9)	4.4 (3.5–5.6)	57 (32–72)	77
Larsen et al. [21]	2020	9	9/9 (100%)	All unruptured	8 (5–23)	NA	60 (48–84)	76
Matsushige et al. [22]	2019	4	4/4 (100%)	Ruptured	6.01 (3–14)	4.1 (3–5)	67.5 (37–91)	83
Quan et al. [19]	2019	9	6/9 (66.7%)	All unruptured	23	16.4	55.4	60
Larsen et al. [23]	2018	13	5/13 (38.5%)	NA	10.3 (6–17)	6.6 (5–9)	53.4 (36–73)	62
Hudson et al. [24]	2018	10	5/10 (50%)	All unruptured	NA	NA	55	90
Shimonaga et al. [25]	2018	9	5/9 (55.6%)	All unruptured	7.2 (6–9)	6.2 (5.3–7)	66.9 (47–81)	44

**Table 2 ijms-25-02700-t002:** A summary of individual study findings. Numbers displayed are *p* values for each parameter investigated. Red denotes the statistical significance presented. Blank values denote the parameter was not investigated. NA indicates where statistical tests were not performed. * Larsen 2020 reported an association between enhancement and a composite endpoint defined as the presence of one or both of vasa vasorum or MPO (*p* value 0.032). They did not report any statistics for individual parameters.

	Thrombus	Wall Thickness	Vasa Vasorum	Macrophage			T Cell	B Cell	Endothelium		Glycans
	H&E/ Masson’s	Microscopy	Microscopy	CD68	MPO	NLRP3	CD3	CD20	CD34	CD31	Alcian Blue
Zhong et al. [20]	0.012	0.124	0.018	0.002	0.001		0.081	0.375	0.018		
Larsen et al. [21]	NA		NA *		NA *				NA		
Matsushige et al. [22]	NA	NA	NA								
Quan et al. [19]				<0.05		<0.05					
Larsen et al. [23]			NA		NA				NA		
Hudson et al. [24]		0.003		0.048							
Shimonaga et al. [25]		NA		NA						NA	NA

## 3. Results and Discussion

### 3.1. Search Results

Searching the databases resulted in 228 publications, of which 216 were unique studies. Abstract searches identified 15 studies that met the inclusion criteria. Searching the bibliographies of the 15 included studies resulted in no further eligible publications. We identified 81 studies that citied the initial 15 studies; 1 of these met the inclusion criteria on abstract screening. A full-text review of the 16 studies resulted in 7 meeting the inclusion criteria.

Two of these studies were case series, while the other five were case-control studies. Six studies included the individual aneurysm data within the paper or Appendix A. 

Only one study (Quan et al. [19]) did not provide individual aneurysm-level data in the paper, Appendix A or a graphical format from which the appropriate summary or individual statistics could be extracted.

The seven eligible studies reporting a series of patients with histological specimens of the aneurysm wall collected at surgery to secure an intracranial aneurysm with accompanying vessel wall imaging are summarised in Table 1. Five of these studies included controls (non-enhancing aneurysms), and two presented the histological results of enhancing aneurysms only. 

### 3.2. Risk of Bias Assessment

Risk of bias was assessed in all studies using the Joanna Briggs Institute checklist for case series or case–control studies as required. The results are displayed in Appendix A.

### 3.3. Pathological Markers Reported

The seven papers identified varied in the choices of pathological markers investigated. In total, 11 different pathological markers were assessed and correlated with vessel-wall imaging. These included CD68, MPO and NLRP3 as markers of pathological macrophage infiltration; CD3 as a marker of T cells; CD20 as a marker of B cells; CD34 as a marker of endothelial cells; and CD31 also as a marker of endothelial cells, although it is expressed in endothelial cells, macrophages and lymphocytes. Vasa vasorum were identified and analysed via light microscopy. Haematoxylin and Eosin (H&E) stains and Alcian blue were used to identify thrombus and mucinous degeneration, respectively. Finally, wall thickness was assessed using light microscopy. A summary of the individual analyses for each of the studies is displayed in Table 2.

### 3.4. Narrative Review of Study Findings

Zhong et al. [20] reported a significant positive association between the presence of thrombi and aneurysm-wall enhancement. Matsushige et al. [22] and Quan et al. [19] both investigated thrombi but did not apply statistical tests to their smaller samples (four and eight, respectively). Pooled univariate analysis of these studies did not quite reach significance (*p* = 0.06).

Macrophages were assessed with CD68, MPO and NLRP3 immunostaining. Both CD68 and MPO were positively associated with enhancement, as reported by Zhong, Larsen 2020 [21], Larsen 2018 [23] and Shimonaga 2018 [25]. Quan et al [19]. reported a positive association for enhancement with CD68 and NLRP3, but the data were not available for pooling. Pooled univariate analysis of available data showed a significant association for both CD68 and MPO with enhancement (*p* = 0.01 and 0.002, respectively).

Only Zhong et al. [20] reported on associations between enhancement and CD3 (T cell) and CD20 (B cell), neither of which was significantly associated with enhancement.

Three studies performed CD34 staining [20,21,23]. All assessed neovessel endothelium within the aneurysm wall. This could not be detected in any of the aneurysms studied by Larsen et al. Zhong et al. reported a positive association between CD34 staining and enhancement (*p* = 0.018). Pooled univariate analysis showed an association between CD34 immunostaining and enhancement (*p* = 0.007). None of the three studies described the endothelial lining of the aneurysm lumen and its integrity.

Both CD31 and Alcian blue were assessed by Shimonaga et al., but no statistical analysis was presented [25]. However, 5/5 enhancing aneurysms stained positive for both, and 4/4 non-enhancing aneurysms did not stain for either. The authors described CD31 as marking neovascularisation (but it is known to be expressed by macrophages and lymphocytes) and Alcian blue as marking mucinous degeneration, with no reference to its staining of glycans.

### 3.5. Meta-Analysis of Predictors of Aneurysm-Wall Enhancement

The results of the meta-analysis of the predictor variables are displayed in Figure 2 as forest plots.

Univariate regression models utilising available aneurysm-level data for unruptured aneurysms (i.e., excluding ruptured aneurysms) were built to inform multivariable logistic regression models accounting for wall thickness (Table 3), which had not been corrected for in the underlying studies and could therefore not be accounted for in the metanalysis. Thrombus, CD68, MPO, CD34 and wall thickness were associated with enhancement during univariate analysis. A multivariate model of these five variables showed no significant relationship with enhancement for any variable (model not presented).

This could be due to small sample size (n = 24), effects in opposing directions or collinearity between variables. The latter was investigated with chi-square and Cramer V analysis, and a correlation matrix was constructed (Figure 3). The analysis suggested that there was a correlation between the predictor variables, with large statistical associations shown by chi-square analysis. In particular, there was a correlation between CD68, MPO and CD34.

Wall thickness also showed an association with four out of the six variables that were associated with vessel-wall enhancement (thrombus, CD68, MPO and CD34, *p* = 0.005, *p* =0.002, *p* = 0.003, *p* = 0.016, respectively, on Kruskal–Wallis rank sum test). Alcian blue and CD31 were not significantly associated with wall thickness (*p* = 0.08 and *p* = 0.09, respectively). 

Therefore, four further models were built (one each for thrombus, CD68, MPO and CD34), with wall thickness as a second predictor for each analysis. CD68 and MPO but not thrombus or CD34 remained significant (Table 3). Sensitivity analysis including ruptured aneurysms yielded similar results, except for thrombus, which was statistically insignificant. Insufficient data were available to account for aneurysm size in a similar manner.

### 3.6. Main Study Findings

In this study, we have shown that macrophage infiltration as determined by immunostaining for CD68 and MPO, neovascularisation as determined by immunostaining for CD34 and CD31, thrombus and thickness of the aneurysm wall were positively associated with aneurysm enhancement on vessel-wall imaging. Additionally, mucinous degeneration (Alcian blue) may be positively associated with enhancement. These markers were all correlated, however, and none were significantly associated according to multivariate regression (albeit with a reduced sample size in a complete case analysis). There were also insufficient data to establish if these were independent of aneurysm size. It is essential that this latter point is resolved, as there is clearly a relationship between aneurysm size and enhancement. If VWI provided no information independent of size, this would bring its clinical use into question, given that aneurysm size can be much more easily derived from MRA, which is routine, fast and requires no contrast.

### 3.7. Determining Causal Mechanisms of Aneurysm Enhancement

There are good mechanistic reasons for any of these parameters to be associated with aneurysmal enhancement, and the question arises as to which of them are causal. Macrophages will be more common in inflamed aneurysm walls, and inflammation is associated with increased enhancement on MRI, although the underlying mechanisms in different structures are not always clear. While parallels with other aneurysm wall imaging techniques such as ferumoxytol-enhanced MRI are attractive, there is no evidence of specific gadolinium uptake by macrophages as has been proposed for ferumoxytol [26]. Therefore, an alternative explanation would be that enhancement is due to an increased volume of water in inflamed tissue, allowing pooling of gadolinium. 

In most situations, enhancement seen on MRI is due to increased vascularity of tissue, as seen in tumours, for example. Increased vascularity in an aneurysm could be due to an increased number of mature vasa vasorum or neovascularisation relating to inflammation or neovascularisation within organising thrombus. This meta-analysis has shown an association between CD34-positive vessels in the aneurysm and enhancement. Although the available studies did not always clearly describe what structures were seen, most infer it was neovascularisation relating to inflammation, and this can be distinguished from vasa vasorum and neovascularisation of thrombus based on the location and differential staining of mature and immature endothelial cells with CD34 [27,28]. The data would therefore be consistent with the hypothesis that this is the cause of enhancement. However, given how thin a typical aneurysm wall is, and the sparsity of vessels in the aneurysm wall even in cases where they are present, it would seem unlikely this vascularity is directly detectable via MRI. 

None of these mechanisms preclude that the markers are not just correlated with each other but also with other unmeasured features which are, in fact, causal. They are, for example, correlated with wall thickness, and wall thickness could reflect changes in the extracellular matrix or smooth muscle of the aneurysm wall, which are both well known to be disordered in aneurysms [29,30]. Since aneurysm walls are thin (typically 0.05–0.3 mm [12]) and at the limit of resolution of even high-resolution MRI (voxel sizes in the included studies ranged 0.5–0.94 mm), partial volume effects could lead to thinner aneurysm walls appearing less intense. It was disappointing to discover, therefore, that there were insufficient data to be able to conclusively resolve whether any of the identified univariate associations are independent of each other.

This work also highlights that other components of the aneurysm wall have not been assessed to date, such as smooth muscle and the extracellular matrix, which make up most of the aneurysm wall. These components can undergo extensive proliferation or remodelling in some aneurysms and hence could be either positively or negatively associated with enhancement, depending on how these changes affect the permeability of the tissue to contrast. Another feature not studied in aneurysms imaged with VWI is the integrity of the endothelial lining of the lumen of the aneurysm (CD34 and CD31 have been used to detail neovascularisation). While this is known to be denuded in many cases, it is plausible that non-enhancing aneurysms represent those with an intact endothelial layer. 

Furthermore, there are several other possible mechanisms underlying aneurysm enhancement that occur outside the aneurysm wall that have not been considered in histopathological studies to date. For example, some studies suggest that the appearance of aneurysm enhancement is due to a lower shear and blood flow at the aneurysm wall that is not adequately compensated by the MR acquisition sequence [16]. Hence, studies to investigate whether gadolinium actually deposits within the walls of enhancing aneurysms would be informative. Another possibility is that contrast does not just permeate through the endothelium but through the entire aneurysm wall and collects in a surrounding layer. This may vary depending on whether an aneurysm is in a free space within the cerebrospinal fluid compartment, adherent to pia mater or protruding into parenchyma. The observation that dynamic contrast-enhanced MRI (DCE) of aneurysms gives distinct and complementary data to VWI might support this concept [31,32], as does the anecdotal observation that a small subset of aneurysms show extremely thick enhancement—thicker than the expected thickness of an aneurysm wall.

### 3.8. Limitations

Although this meta-analysis followed a robust methodology, it has a number of limitations, many of which relate to the availability of data in the literature. These limitations relate to the number of cases and the depth of phenotyping. Overall, there were only seven papers describing 81 aneurysms, and data could be extracted for only 73 aneurysms. The overwhelming majority of cases were unruptured, with little information available for ruptured cases in which the cause of enhancement may differ (for example, by collecting within the thrombus at the rupture point). It is also inevitable, due to the nature of histological examination, that cases were biased towards those needing surgery. This will have meant that the aneurysms studied were larger, and patients were more likely to have had greater risk factors for rupture than average. This is consistent with the fact that 68% of aneurysms showed enhancement—much higher than in unselected series of unruptured intracranial aneurysms [33]. Furthermore, most series did not report whether they represented consecutive cases or not, and what criteria would have triggered preoperative VWI or surgical excision of the aneurysm. None of the studies measured all histological parameters, and only one undertook more than half of them.

Due to these limitations in data availability in the literature, it has not been possible to establish whether predictors were independent of each other or not, and it cannot be ruled out that other predictors were not identified due to lack of power. All these selection biases in individual studies that have small sample sizes also increase the risk of publication bias. Unfortunately, due to the low number of comparable publications within this meta-analysis, it was not possible to make a formal assessment of heterogeneity or publication bias. 

There are further limitations in terms of how both VWI and histopathology were quantified. VWI sequences, protocols and contrast agents varied between studies, and although all used enhancement as a binary variable, this is inherently subjective, and it may be better quantified. Similarly, all histological features were binarised as present or absent, which may oversimplify the underlying processes as well as ignoring heterogeneity between different regions of the aneurysm dome. There were also limitations to how specific features such as wall thickness were measured. For example, measurement of wall thickness on 2D histology is flawed as it is highly dependent on slice orientation, which can be difficult with very thin aneurysm walls, which lose their spatial characteristics when they collapse and during histological preparation; robust thickness measurement therefore requires 3D techniques such as 3D histology reconstruction or micro-CT [34,35]. However, possibly the biggest and most remediable limitation was the lack of information on aneurysm size, precluding analysis to determine if any associations are independent of aneurysm size.

## 4. Conclusions

The current literature suggests that aneurysm enhancement is associated with inflammatory infiltrates, neovascularisation and wall thickness. The body of published evidence is small, with only 81 aneurysms being subjected to pathological analysis and significant heterogeneity of studies. It is therefore not possible to determine if these histological parameters are independent of each other or of aneurysm size and which is most likely to be causally related to enhancement. Moreover, many other potential underlying mechanisms of aneurysm wall enhancement exist. More studies are required in order to clarify the pathological basis of aneurysm enhancement on VWI. 

## Figures and Tables

**Figure 1 ijms-25-02700-f001:**
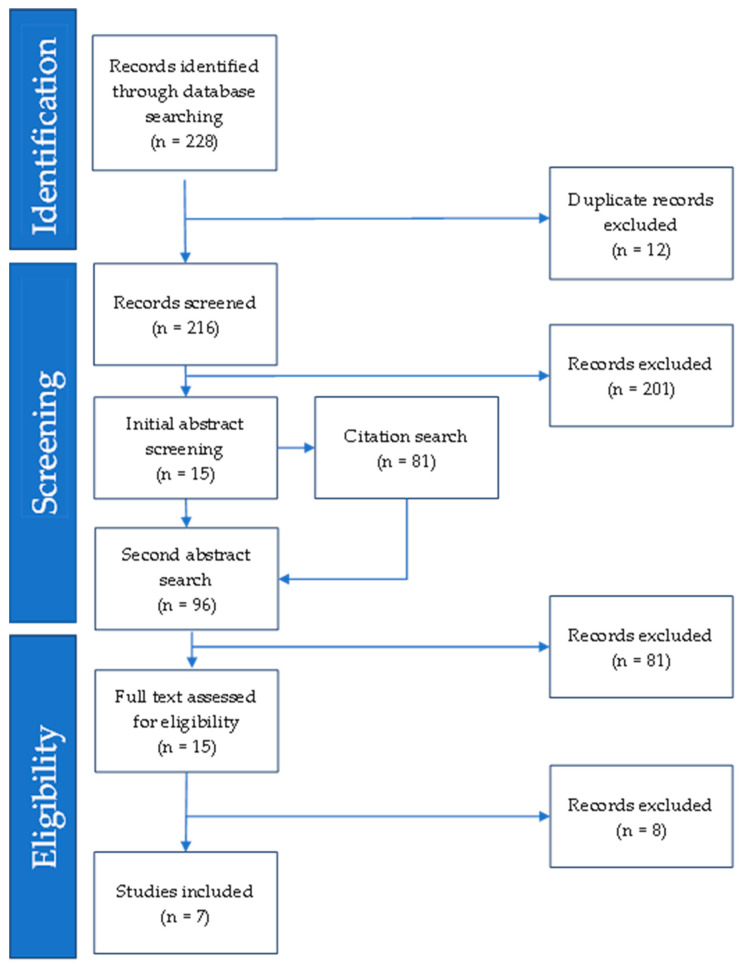
Flow diagram of the search strategy and study selection.

**Figure 2 ijms-25-02700-f002:**
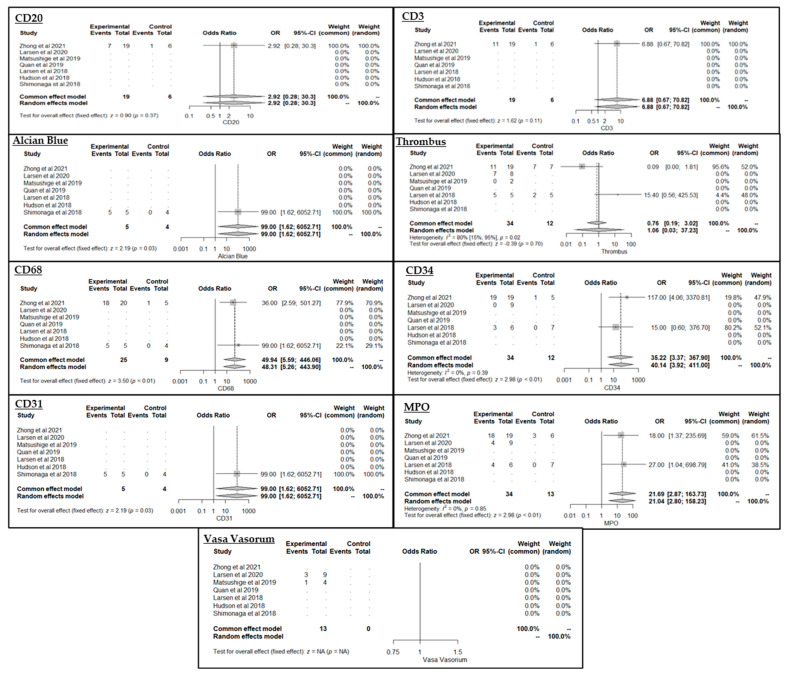
Forest plots of markers of enhancement. CD68, MPO, CD34, CD31 and Alcian blue were all significantly associated with enhancement when pooled. Although Alcian blue and CD31 were statistically significant, their odds ratio is extremely large, owing to the uncertainty of the data, as both only had 9 observations for each. Thrombus, vasa vasorum, CD3 and CD20 were not statistically significant. Thrombus and vasa vasorum included 4 ruptured aneurysms in their analysis [19,20,21,22,23,24,25]. Where the data is not available NA is shown.

**Figure 3 ijms-25-02700-f003:**
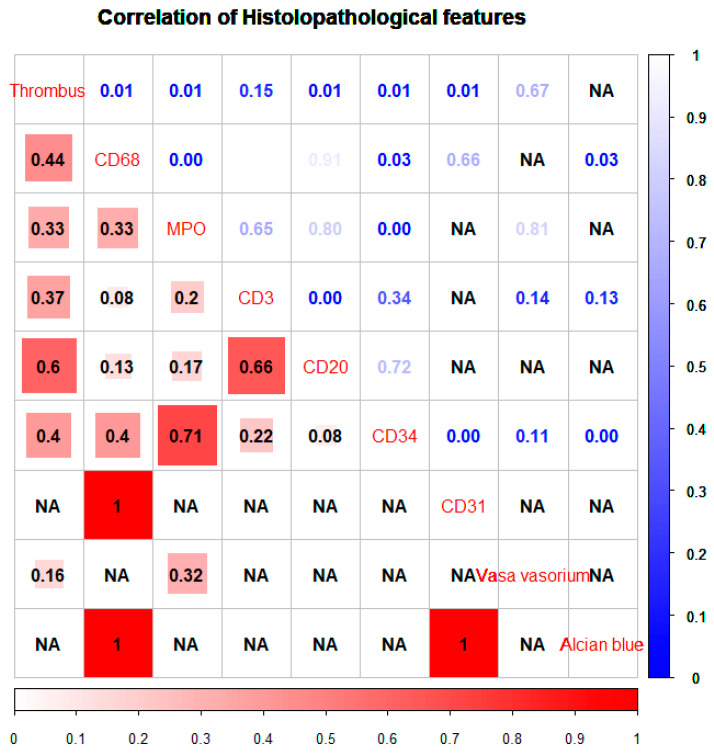
A correlation matrix of histological features. The Cramer V association is shown in the lower left in red. The associated *p* value from a X^2^ test is shown in blue on the top right of the matrix. NA is shown where there are no data that cross over to assess correlation.

**Table 3 ijms-25-02700-t003:** Pooled univariate and multivariate analysis modelling excluding the ruptured aneurysms. Each line of the multivariable section represents a separate result for a model analysing the pathological marker accounting for aneurysm size as a second covariate. * Denotes statistical significance with a *p* value < 0.05. NA denotes insufficient data to correctly quantify results due to either a lack of data or data that were only enhancing or non-enhancing.

Univariate	Odds Ratio	Confidence Interval (95%)	*p* Value	Number of Observations
Thrombus	6.00	1.32–43.16	0.035 *	45
CD68	92.0	10.44–2303.64	0.001 *	34
MPO	10.0	2.73–49.06	0.001 *	57
CD3	6.88	0.88–145.83	0.105	25
CD20	2.92	0.36–61.98	0.37	25
CD34	20.7	3.66–392.43	0.005 *	56
CD31	NA	NA	NA	9
Vasa vasorum	NA	NA	NA	13
Alcian blue	NA	NA	NA	9
Wall thickness	1.01	1.00–1.02	0.020 *	46
**Multivariate** **(Accounting for** **wall thickness)**	**Odds Ratio**	**Confidence** **interval (95%)**	***p* value**	**Number of** **observations**
Thrombus	3.76	0.601–32.13	0.174	36
CD68	112.09	9.02–1.36 × 10^4^	0.003 *	34
MPO	1740.373	36.5–2.17 × 10^7^	0.005 *	35
CD34	NA	NA	1	34

## Data Availability

The datasets used and/or analysed during the current study are available from the corresponding author on reasonable request.

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
