# Peer review of "A Systematic Review and Meta-Analysis of the Pathology Underlying Aneurysm Enhancement on Vessel Wall Imaging"

_ijms, 2024, doi:10.3390/ijms25052700_

Round 1

Reviewer 1 Report

Comments and Suggestions for Authors

This paper reports on a meta-analysis and systematic review of pathology that may underlie enhancement of aneurysm walls seen during contrasted vessel wall imaging on MRI. Based on 7 eligible studies of 72 our of 82 aneurysms,  univariate analysis showed macrophage markers CD68 and MPO, endothelial cell markers CD34 and CD31, glycans Alcian blue and wall thickness were positively associated with enhancement. The number of aneurysms analysed and parameters considered was small - still aneurysm enhancement was felt to be associated with inflammatory infiltrate and neovascularisation. While a well-done summary, the small numbers and other limitations mentioned by the authors precludes drawing conclusions impacting clinical practice

The authors may wish attend to the following:

1.       page 2 - Fig 1 – it appears to be cut off

2.       Methods (major) – please spell out clearly the inclusion and exclusion criteria

3.       Table 1 – age – in years?

4.       Table 2 – what does ‘NA’ stand for? As a blank space means the parameter was not investigated

5.       Page 9 – was the larger font of the last sentence intentional?

Author Response

This paper reports on a meta-analysis and systematic review of pathology that may underlie

enhancement of aneurysm walls seen during contrasted vessel wall imaging on MRI. Based on 7

eligible studies of 72 our of 82 aneurysms, univariate analysis showed macrophage markers CD68

and MPO, endothelial cell markers CD34 and CD31, glycans Alcian blue and wall thickness were

positively associated with enhancement. The number of aneurysms analysed and parameters

considered was small - still aneurysm enhancement was felt to be associated with inflammatory

infiltrate and neovascularisation. While a well-done summary, the small numbers and other

limitations mentioned by the authors precludes drawing conclusions impacting clinical practice

The authors may wish attend to the following:

  1. page 2 - Fig 1 – it appears to be cut off

We apologise for this. We have reformatted it. Please let us know if the formatting problems persist when after it has been uploaded to the site.

  1. Methods (major) – please spell out clearly the inclusion and exclusion criteria

We have now clarified the inclusion and exclusion criteria in the section describing the literature search:

“We included cohort studies, case-control studies and case series which reported on at least one histological marker from a saccular intracranial aneurysm wall obtained at surgery that had been imaged preoperatively with MRI Vessel Wall Imaging at 1.5 or 3 Tesla and were written in English. Case reports, studies that reported on intracranial aneurysms thought to be of an immune related aetiology, blister aneurysms and thrombosed aneurysms were excluded.”

  1. Table 1 – age – in years?

We have added “years” in brackets next to the title.

  1. Table 2 – what does ‘NA’ stand for? As a blank space means the parameter was not

Investigated.

A blank means that the parameter was not measured

A NA means that no statistical test was applied in the published manuscript associating it to vessel wall imaging.

We have added a description of this to the table legend

“Blank values denote the parameter was not investigated. NA indicates where statistical tests were not performed.“

  1. Page 9 – was the larger font of the last sentence intentional?

We apologise for this is formatting error that seems to have occurred after transfer to the manuscript template style and has been rectified.

Reviewer 2 Report

Comments and Suggestions for Authors

The authors provide an interesting review about vessel wall enhancement imaging for intracranial aneurysm and correlation to histopathological study. 

The work is well described, limitation are clearly stated. Minor remarks can be given to improve the manuscript. 

Could you provide with the Prisma checklist in the supplementary material file? 

Could you provide which author(s) (if multiple specify if it is independentlyand how issue were resolved) performed the literature screening 

Did you try to contact the author of the study (citation 24) to obtain the data ? 

The Figure 1 did not display adequately in the files ? 

In table 1, provide dispersion index as interquartile range for mean size of aneurysm, age…

Please provide “easy to see” Title for each forest plot in Figure 2 

I am not sure to see how the multivariate model is interesting due to the very small amount of data. Including both ruptured and unruptured is a mistake in my opinion and include associated variable (as MPO and CD68). I would only provide univariate analysis. I would specify in the text which data include ruptured and unruptured aneurysm at each time. 

Please support this statement with available literature: “The observation that dynamic contrast enhanced MRI (DCE) of aneurysms gives distinct and complimentary data to VWI might support that concept, as does the anecdotal observation that a small subset of aneurysms show extremely thick enhancement - thicker than the expected thickness of an aneurysm wall. “

Author Response

The authors provide an interesting review about vessel wall enhancement imaging for intracranial aneurysm and correlation to histopathological study. 

The work is well described, limitation are clearly stated. Minor remarks can be given to improve the manuscript. 

Could you provide with the Prisma checklist in the supplementary material file? 

This was included within the supplementary file on submission and where to find them is referenced on the final page of the main manuscript:

“Declarations

Supplemental Material: Supplemental table 1; Supplemental figures 1-2; PRISMA checklist

Please let us know if for any reason these are not available to you.

Could you provide which author(s) (if multiple specify if it is independently and how issue were resolved) performed the literature screening 

A single author performed the screening and search – this author (RD) is now identified within the search section (section 2.1, page 3)

Did you try to contact the author of the study (citation 24) to obtain the data ? 

We did not contact the authors of this study. It was a decision at the start of the study to not contact individual groups for unpublished data as from experience it can be very difficult to get multiple groups to data share particularly in a new area of research where only small series are available as in this case. If we had, it would have increased the number of available aneurysm data from 72 to 81. This is unlikely to significantly alter the findings and would still show the paucity of data within the literature.

The Figure 1 did not display adequately in the files ? 

We apologize for this. We have reformatted it. Please let us know if the formatting problems persist when after it has been uploaded to the site.

In table 1, provide dispersion index as interquartile range for mean size of aneurysm, age…

We have added range as a measure of dispersion. Not all studies provide individual patient level data for all variables. Some only provide ranges for variables like age precluding presentation of interquartile range.

Please provide “easy to see” Title for each forest plot in Figure 2 

We thank the reviewer for pointing out that it was difficult to see the text on the x axis indicating what each individual panel in the Forrest plot referred to. To make this clearer we have added a title in bold with the marker in the panel to the top left of each plot.

I am not sure to see how the multivariate model is interesting due to the very small amount of data. Including both ruptured and unruptured is a mistake in my opinion and include associated variable (as MPO and CD68). I would only provide univariate analysis. I would specify in the text which data include ruptured and unruptured aneurysm at each time. 

One of the main reasons for undertaking the study was to assess if the reported predictors of enhancement are independent of wall thickness and aneurysm size. This could only be achieved with a multivariate analysis as the underlying studies did not present their results corrected for these variables and the main metanalysis forest plots could not be corrected for these. The only purpose of the univariate analysis was, therefore, to inform the multivariate analysis so that wall thickness could be accounted for and independence of the predictors tested. Without the multivariate analysis, the univariate analysis would be providing little more than the main study level metanalysis therefore (one would be a study level and the other an aneurysm level analysis. Therefore, if the multivariate analysis was to be removed the univariate should be removed.

We feel there is value, however, even if negative to attempt to examine independence of the predictors. An equivocal result due to lack of data is still informative to highlight that insufficient data exists and it therefore requires further study.

We have therefore a) modified the analysis as suggested by the reviewer and b) explained its purpose more clearly. We have excluded ruptured aneurysms from both uni and multivariable analyses as suggested. We have also built four additional models (thrombus+wall thickness, CD68+wall thickness, CD34+wall thickness, and MPO+wall thickness) to specifically look to see if these markers are independent of wall thickness and have updated the text as:

Univariablee regression models utilizing available aneurysm level data for unruptured aneurysms (ie excluding ruptured aneurysms) were built to inform multivariable logistic regression models accounting for wall thickness (Table 3), which had not been corrected for in the underlying studies and could therefore not accounted for in the metanalysis. Thrombus, CD68, MPO, CD34 and wall thickness were associated with enhancement on univariate analysis. A multivariable model of these 5 variables showed no significant relationships with enhancement for any variable (model not presented).

This could be due to small sample size (n=24), effects in opposing directions or collinearity between variables. The latter was investigated with chi square and Cramer V analysis, and a correlation matrix was constructed (Figure 3). The analysis suggested that there was correlation between the predictor variables with large statistical associations shown by chi square analysis. In particular there was correlation between CD68, MPO and CD34.

Wall thickness also showed an association with 4 out of the 6 variables that were associated with vessel wall enhancement (Thrombus,CD68, MPO and CD34,p=0.005, p =0.002, p=0.003, p=0.016, respectively on Kruskal-Wallis rank sum test). Alcian blue and CD31 were not significantly associated with wall thickness (p=0.08 and p=0.09 respectively).

Therefore, four further models were built with one for each of Thrombus, CD68, MPO, CD34 with wall thickness as a second predictor for each analysis. CD68 and MPO but not Thrombus or CD34 remained significant (Table 3). Insufficient data was available to account for aneurysm size in a similar manner.”

Please support this statement with available literature: “The observation that dynamic contrast enhanced MRI (DCE) of aneurysms gives distinct and complimentary data to VWI might support that concept, as does the anecdotal observation that a small subset of aneurysms show extremely thick enhancement - thicker than the expected thickness of an aneurysm wall.

We have added the two relevant references showing that DCE might offer complimentary data to VWI to this sentence in the manuscript.

Reviewer 3 Report

Comments and Suggestions for Authors

This study provides a systematic review enhanced intracranial aneurysm that underwent vessel wall imaging. The authors found that macrophage and endothelial cell markers, glycans and vessel wall thickness were positively associated with enhancement. However, the underlying data was insufficient for further conclusions.

Overall, the study deals with an interesting and recent topic that is heavily debated in the community. The methods are carefully carried out and the authors are self-critical with their findings. The following remarks remain:

-          The provided PDF file of the manuscript has a formatting issue of Figure 1 (mainly disappears on the bottom of page 2). Hence, I cannot evaluate it.

-          From 228 studies only 7 met the inclusion criteria and were selected for further analysis. In total, 73 intracranial aneurysms were evaluated. Since this review claims to be a meta-analysis, the number of studies (and cases) considered is rather low. Hence, the relevance of the topic might be limited.

-          The limitation section is clear and honest and contains all relevant constraints. Unfortunately, the corresponding data situation leads to an unsatisfactory study outcome in terms of added knowledge. Therefore, it is rather suggested to shift the focus of the study and use more robust and homogenized data.

-          The list of authors appears to be quite long for this kind of contribution. The “Authors’ contributions” states: JY,LKM,NB,OLK,JM.AD,AN,AD,ROC,MF,IG were a major contributor in writing the manuscript. It is hardly believable that 11 professionals contributed to the writing process.

-          The list of references (n=35) is relatively low for a review article

Author Response

This study provides a systematic review enhanced intracranial aneurysm that underwent vessel wall imaging. The authors found that macrophage and endothelial cell markers, glycans and vessel wall thickness were positively associated with enhancement. However, the underlying data was insufficient for further conclusions.

Overall, the study deals with an interesting and recent topic that is heavily debated in the community. The methods are carefully carried out and the authors are self-critical with their findings. The following remarks remain:

-          The provided PDF file of the manuscript has a formatting issue of Figure 1 (mainly disappears on the bottom of page 2). Hence, I cannot evaluate it.

We apologize for this. We have reformatted it. Please let us know if the formatting problems persist when after it has been uploaded to the site.

-          From 228 studies only 7 met the inclusion criteria and were selected for further analysis. In total, 73 intracranial aneurysms were evaluated. Since this review claims to be a meta-analysis, the number of studies (and cases) considered is rather low. Hence, the relevance of the topic might be limited.

We agree the number of studies and cases is low. However, all studies with the required histopathologic analysis and concurrent MRI VWI findings were considered and this reflects the available literature. There is no minimum number of studies or cases for a metanalysis. Indeed the purpose of metanalysis to increase the number available for analysis to overcome the limitations of available in individual studies. We would argue the low numbers if anything makes it more important to summate all the available data to get to a dataset that you might feasibly be able to conclude something from.

We find it striking how confident people at meetings are what VWI represents and how much of the literature reads as if it is already known what aneurysm enhancement is due to, when we think there is very little data to confidently conclude this (and presumably the reviewer agrees as they have pointed out the paucity of data available). This summation of the available data should be of interest to anyone interested in imaging and prediction of rupture of aneurysms.

-          The limitation section is clear and honest and contains all relevant constraints. Unfortunately, the corresponding data situation leads to an unsatisfactory study outcome in terms of added knowledge. Therefore, it is rather suggested to shift the focus of the study and use more robust and homogenized data.

We have removed the cases of ruptured aneurysms from the multivariable model to make it a more homogeneous population as the reviewer suggested.

-          The list of authors appears to be quite long for this kind of contribution. The “Authors’ contributions” states: JY,LKM,NB,OLK,JM.AD,AN,AD,ROC,MF,IG were a major contributor in writing the manuscript. It is hardly believable that 11 professionals contributed to the writing process.

      The conceptualisation of this study came from our study group which meets monthly. The group is undertaking its own research study in this area and this was its attempt to understand the available literature. While the main literature search was the work of one author (RD) and the analysis done in close conjunction with another (DB) the interpretation and contextualisation of these are highly multidisciplinary and required input from our whole study group. In order to move the field forwards future studies will need to use a wide range of techniques. No individual author could feasibly have the required expertise across all these disciplines. These areas of expertise are:

      RD, NB, DB – neurosurgery (anatomy, treatment and natural history of aneurysms)

      JY, MK – neuropathology (light microscopy and histopathology)

      KA, LM – endothelial imaging

      RD – electron microscopy

      OK – micro CT (for aneurysm thickness measurement)

      JM, AD, AN – neuroradiology (for interpretation and measurement of VWI)

      AD – Medical physics (for MRI VWI protocol development)

      RC – Neuroanatomy of cerebral vessels

IG – Experimental neurology (expertise in the complimentary technique of DCE)

-          The list of references (n=35) is relatively low for a review article

We have added two references.

Otherwise, the number of references reflects this is a new and novel area of research in which limited amounts of data is available and we believe adding further references would not be directly relevant to the topic discussed.